# Influence Mechanism of Cu Layer Thickness on Photoelectric Properties of IWO/Cu/IWO Films

**DOI:** 10.3390/ma13010113

**Published:** 2019-12-25

**Authors:** Fengbo Han, Wenyuan Zhao, Ran Bi, Feng Tian, Yadan Li, Chuantao Zheng, Yiding Wang

**Affiliations:** State Key Laboratory of Integrated Optoelectronics, College of Electronic Science and Engineering, Jilin University, 2699 Qianjin Street, Changchun 130012, China; fbhan521@163.com (F.H.); zhaowy18@mails.jlu.edu.cn (W.Z.); ranbi18@mails.jlu.edu.cn (R.B.); tianfeng17@mails.jlu.edu.cn (F.T.); yadancomeon@163.com (Y.L.)

**Keywords:** multilayer, thin film, transparent conductive

## Abstract

Transparent conductive IWO/Cu/IWO (W-doped In_2_O_3_) films were deposited on quartz substrates by magnetron sputtering of IWO and Cu in the Ar atmosphere. The X-ray diffraction (XRD) patterns identified the cubic iron–manganese ore crystal structure of the IWO layers. The influence of the thickness of the intermediate ultra-thin Cu layers on the optical and electrical properties of the multilayer films was analyzed. As the Cu layer thickness increases from 4 to 10 nm, the multilayer resistivity gradually decreases to 4.5 × 10^−4^ Ω·cm, and the optical transmittance in the mid-infrared range increases first and then decreases with a maximum of 72%, which serves as an excellent candidate for the mid-infrared transparent electrode.

## 1. Introduction

Transparent conductive oxides (TCOs) are intrinsic n-type semiconductors with an inherent natural donor of oxygen vacancies and additional external doping of interstitial atoms [1,2]. TCOs usually consist of transition metal cations (called TCO cations) because the powerful delocalized s orbitals of these cations form a dispersed conduction band. Their effective electron mass is very small, which assures high mobility and a wide band gap [3], such as In^3+^ [4,5], Zn^2+^ [6,7,8], and Sn^4+^ [9,10]. Due to the remarkable optical and electrical properties, doped TCO layers are integrated into many optoelectronic devices, which are widely used in photovoltaic solar cells [11,12,13], flat panel displays [14,15,16], photoelectric sensors [17], and film transistors [18,19,20]. However, it is still a big challenge to optimize the optoelectronic properties of the film by balancing the excellent electrical conductivity and high optical transmittance to maintain the extremely small optical absorption and reflection and high carrier concentration [21]. In general, the transparency of the film can be significantly increased by decreasing the carrier concentration, which sacrifices the conductivity of the film. However, since the conductivity is proportional to the product of carrier mobility and concentration, the conductivity can remain unchanged by increasing the carrier mobility but reducing the carrier concentration, which extends the transparent window to the infrared wavelength [22]. It is established that the substitution of high-valent dopant for ions in the crystal lattice can provide extra electrons, which reduces the transmittance of the film. As the doping level decreases, the film can achieve relatively high transparency in the infrared range.

In recent years, the price of indium has been rising constantly, resulting in a huge increase in cost. The disadvantages of traditional TCO films become more and more obvious [23]. Single-layer TCO films can no longer meet needs, and the development of a novel film is imminent. The well-developed techniques of depositing metal layers and semiconductor films lead to further advances and opportunities in this field. An extremely thin metal layer can be added between dielectric layers in order to increase effective conductivity [24]. The multilayer film also has a smaller thickness compared to the single one [25,26]. Some multilayer films composed of a variety of TCO and metals have been reported using different high-temperature deposition methods, such as electron beam evaporation [27,28], sputtering [29], chemical vapor deposition [30], and pulsed laser deposition [31]. The magnetron sputtering method is a valid technique that can produce films with reasonable quality at a stable deposition rate. However, mid-infrared photoelectric properties of multilayer films deposited by magnetron sputtering at room temperature have rarely been reported. Therefore, the IWO/metal/IWO multilayer film was proposed for operation in the mid-infrared range. The IWO/Cu/IWO sandwich-type film was fabricated by radio frequency (RF) and direct current (DC) magnetron sputtering at room temperature. Cu was deposited as an intermediate metal layer by DC magnetron sputtering in order to improve the conductivity. Tungsten-doped indium oxide films were deposited on both sides of the Cu layer by RF magnetron sputtering, which was used to suppress reflectance from the metal layer in the mid-infrared region. The photoelectric properties of the IWO/Cu/IWO multilayer films were studied as a function of Cu layer thickness in order to obtain superior mid-infrared TCO films at room temperature.

## 2. Materials and Methods

Multilayer IWO/Cu/IWO thin films were deposited on quartz and sapphire substrates of 0.5 mm thickness, using the sp-203 magnetron sputtering system manufactured by LJ company in Taiwan. The quartz substrate was ultrasonically cleaned in acetone and ethanol solution for 20 min to remove organic impurities on the surface, then rinsed in deionized water to remove the solvent residue, and finally dried by pure nitrogen. The sputtering system contained a ceramic target of In_2_O_3_/WO_3_ (97.5/2.5 wt.%), a Cu metal target (99.99% of purity), and a tray that was rotatable under the target at a speed of up to 20 r/min to ensure uniform film deposition. The distance between the sputtering target and the rotating tray was set to 14 cm. The multilayer film was prepared under the chamber base pressure of less than 5 × 10^−5^ Pa. The IWO and Cu layers were deposited with a constant power of 35 W and 50 W, respectively. Relatively low-power sputter deposition was used to avoid surface damage caused by high energy ions. The target was pre-sputtered for 30 min to remove surface impurities. For the sandwich structure, the upper and bottom IWO layers were deposited by RF (13.56 MHz) magnetron sputtering under a pressure of 0.2 Pa of high purity Ar atmosphere with 25 sccm at room temperature, and an intermediate Cu layer was prepared by DC magnetron sputtering. The sputtering parameters for the IWO and Cu layers are listed in Table 1. The growth rate of the Cu layer was 4 nm/min, whereas the growth rate of the IWO layers was as low as 0.8 nm/min due to the low sputtering power. The deposition time scale was used to control the film thickness. The sputtering time of the IWO film was set to 50 min at a constant (40 nm), and the Cu layer sputtering time was set to 0, 1, 1.25, 1.5, 2, and 2.5 min (0, 4, 5, 6, 8, and 10 nm). Here, different IWO films were grown on the substrate, whose thicknesses were determined by sputtering time. With a step profiler, the measurement results of more than 10 groups of samples prove that the deposition rate can be accurately controlled with an error of <5%. The growth rate of the Cu layer on the IWO film was also stable. Finally, the obtained samples were annealed at 350 °C for good electrical performances.

The surface morphology of the metal layers was studied by UItima IV atomic force microscopy (AFM) manufactured by Japanese Science Corporation (Tokyo, Japan). The infrared transmission spectrum was measured in a wavelength range of 2.5–5 μm using a Nicolet iS50 Fourier infrared spectrometer (FTIR) manufactured by Thermo Fisher Scientific (Waltham, MA, USA). The crystallinity of the films was analyzed by a TTRIII combined multifunction X-ray diffractometer (XRD, Tokyo, Japan). The thickness of the films was measured by a XP-2 step profiler (AMBIOS Technology Corporation, Milpitas, CA, USA). The electrical properties of the films were characterized by a SKP5050 Kelvin Probe and Hall Effect Tester (KP Technology Ltd., Wick, United Kingdom), which were used to measure the work function or surface potential of materials and to test the Hall effect and magnetoresistance characteristics.

## 3. Results and Discussion

### 3.1. Morphology and Structure

The surface morphology of the evolution progress obtained by atomic force micrographs (AFM) of Cu layers prepared on IWO films by RF magnetron sputtering are shown in Figure 1. The root-mean-square surface roughness of Cu layers was found to be (a) 1.36 nm, (b) 1.28 nm, and (c) 2.12 nm, respectively. The growth morphology of the Cu layer was systematically studied in order to evaluate the influence of the intermediate Cu layer on the photoelectric properties of the multilayer films. The evolution of the Cu layer is concluded with the following processes: separated Cu islands, mixed Cu islands, and semi-continuous Cu layers. Figure 1a,b show the structure of the Cu island, which can form a semi-continuous layer as it reaches the critical thickness, as shown in Figure 1c. There is evident variation in root-mean-square surface roughness, which declines from 1.36 nm to 1.28 nm for the Cu layers with 4 nm and 6 nm thicknesses. This can be a result of the merger of smaller islands. As the Cu layer thickness is increased to 8 nm, the film shows great undulations and the roughness rises to 2.12 nm.

Figure 2 shows the surface morphology images of the bare IWO (80 nm thickness) thin film with a roughness of 0.27 nm, the IWO film with a roughness of 0.23 nm on a sapphire, and a multilayer film with a 6 nm Cu layer with a surface roughness of 1.29 nm. This shows that the substrate does not greatly affect the surface morphology of the IWO film. The surface topography in Figure 2b indicates that the bottom IWO film reveals a uniform and smooth surface, which is critical for growing the Cu layer with a good surface topography. However, the surface roughness of the IWO/Cu/IWO film in Figure 2c is large because of the large roughness of the intermediate Cu layer, which affects the growth of the upper IWO film. It is apparent that the morphology of the Cu layer affects the performance of the multilayer film.

Figure 3 shows the XRD patterns of the IWO/Cu/IWO thin films prepared on quartz substrates at room temperature with different thicknesses of Cu layers. The peaks at around 31° and 36° are consistent with the (222) and (400) diffraction peaks of cubic bixbyite In_2_O_3_ structure [32], which illustrates that tungsten doping does not notably change the crystal structure of indium oxide [33]. The diffraction peaks of the Cu layer are hard to detect since they should be significantly broadened with such a small thickness. However, within the resolution threshold, the characteristic Cu (111) peak can still be identified, although it is unobvious. Insignificant diffraction peaks were observed in the XRD pattern at a thickness of 4, 5, and 6 nm Cu, indicating that the intermediate Cu layers are amorphous, which is an island shape. For the Cu thickness of 8 nm and 10 nm, the XRD peak at 2 theta = 43.3° was noticed as corresponding to the (111) of the positions for Cu diffraction standard, the weak intensity of the diffraction peak revealing that the film quality of the Cu layer is poor, and that the Cu layers are in an approximately semi-continuous state.

### 3.2. Optical Properties

Figure 4 shows the transmission spectra in the mid-infrared region (2.5–5 μm) obtained from the deposited multilayer films on different substrates. It is evident that the transmittance of the multilayer structure is lower than that of the bare IWO film due to the embedded Cu layer, reaching a maximum of 72%. As shown in Figure 4a, the absorption peak at 2.73 μm is due to the hydroxyl impurity inside the quartz substrate. The transmission spectrum of the IWO thin film on the quartz substrate was corrected by that of the uncovered quartz substrate, and we then obtained the transmission spectrum of the bare IWO thin film without the absorption peak at 2.73 μm. With the increase in Cu layer thickness from 4 nm to 6 nm, the transmittance increases first and then decreases in the region of 2.5–3.75 μm. The maximum transmittance of 67% in 2.5–5 μm is attained on IWO/Cu/IWO multilayer films with a 5 nm-thick Cu layer. Figure 4b shows the transmission spectra of IWO/Cu/IWO films on sapphire substrates in the mid-infrared region (2.5–5 μm). It is clear that the transmission also increases first and then decreases. It is obvious that the thickness of the Cu layer plays a crucial role since the IWO layer thickness is unchanged. The evolution of the intermediate Cu layer deposited on the IWO layer follows an island or Volmer–Weber growth mode, which is demonstrated by many metal systems as it grows on the surface of the substrate [34]. In this growth mode, the metal particles first form small clusters around several nuclei, and then grow into islands, which are further combined into a continuous porous film and finally coalesce into a dense film, as illustrated in Figure 1. As the thickness of the embedded metal layer is less than 8 nm, its morphology tends to be discontinuous clusters or islands, like Figure 5a. At this stage, the interplay between the conducting electrons and the surface of the cluster becomes significant due to the extra collision process affecting the effective mean free path, and now depends on the cluster size. This results in the correction of the imaginary part of the bulk metal dielectric function (*ε*_R_) with the following formula [35]:*ε*_R_ = *ε*_m_ + *C/R*(1)
where *ε*_m_ denotes the dielectric constant of the bulk metal, *C* is a constant at a particular wavelength, and *R* depicts the particle radius. It can be obtained from Equation (1) that the transmittance of the metal island rises with the radius of the metal layer increases to a certain extent, with a metal layer thickness of no more than 6 nm. When the thickness of the embedded metal layer is >8 nm, the nucleated clusters or islands begin to aggregate and then join into a sheet, and finally form a porous semi-continuous film, as shown in Figure 5b. The optical transmission depends on the imaginary part of the dielectric function, which is given by the following formula [36,37,38]:*ε*(*ω*) = *ε*_1_(*ω*) + *iε*_2_(*ω*) = (*n*^2^ − *k*^2^) + *i*(2*nk*)(2)
where *n* stands for refractive index and *k* is extinction coefficient. The relationship between the extinction coefficient k and the absorption coefficient of light *η*_abs_ can be written as follows [39]:*η*_abs_ = 2*ωk*/*c*(3)
where *c* is the speed of light in the vacuum. The coefficient *ƞ*_abs_ represents the energy loss ratio of the electromagnetic wave transmitted through the dielectric material, which increases as the thickness of the metal layer rises from 6 nm. Therefore, the transmittance of the IWO/Cu/IWO films on quartz substrate is gradually reduced.

### 3.3. Electrical Properties

Figure 6 reveals the variation in the carrier concentration, mobility, resistivity, and sheet resistance of the IWO/Cu/IWO thin films as a function of Cu layer thickness. The morphology of the copper layer is related to the surface morphology and roughness of the underlying IWO film, which directly affects the electrical properties of the multilayer film. We performed each group of experiment more than twenty times to determine the error of the electrical properties of the thin film by analyzing the obtained carrier concentration, mobility, and resistivity. Although the results obtained are slightly different due to the deviation of Cu, the error is <6%. The resistivity of the bare IWO thin film is 9.86 × 10^−3^ Ω·cm, which is basically the same as that of the multilayer film without a Cu layer. Figure 6a shows the relationship of carrier concentration and Hall mobility of IWO/Cu/IWO films. With increasing Cu layer thickness, an obvious improvement in Hall mobility is observed from 25.8 cm^2^/Vs at 0 nm to 32.1 cm^2^/Vs at 10 nm. The carrier concentration is increased significantly from 2.52 × 10^19^ to 4.28 × 10^20^ cm^−3^, which depends on the increase of the thickness of the Cu layer. Hence, the conduction of Cu is dominant in this multilayer system. As shown in Figure 6a, for the IWO film without a Cu layer, the carrier concentration is 2.52 × 10^19^ cm^−3^ only. Furthermore, with a Cu layer, the carrier concentration rises sharply. Within the range of the Cu layer thickness from 4 to 6 nm, the carrier concentration of IWO/Cu/IWO films rises softly; but, when the thickness is >8 nm, the further increase of the metal layer thickness leads to a huge rise of carrier concentration. When the thickness of the Cu layer is from 4 nm to 6 nm, the carrier concentration of the multilayer film is mainly dominated by the lower and upper lays of IWO, because the deposition of Cu particles mainly exists in an island form and the gap between the islands is relatively large. Free electron transport needs a special quantum tunnel between the islands [40]. The passage of this special tunnel requires electrons to have a tremendous activation energy, and the IWO film has a weak adsorption capacity for the metal layer. Under this condition, the metal layer can provide less free electrons to move. When the thickness is >8 nm, as the islands grow larger, the island gap becomes smaller and the islands begin to merge at an extensive scale. The carrier concentration of the film at this stage will rise sharply. The mobility of the carriers is affected by many scattering factors, such as grain boundary scattering, ionization scattering, interface scattering, etc. However, it is mainly affected by island boundary scattering. The migration rate increases with the size of the metal island, and the relationship between carrier mobility and material scattering is given by [41]:(4)μg=Sq12πm∗kBT12exp−ΦBkT
where *S* is island size, *q* is electronic charge, *Φ*_B_ is the number of potential island boundaries, *k*_B_ is the Boltzmann constant, and *m** is the effective mass of the carrier.

Figure 6b shows the volume resistivity and sheet resistance of IWO/Cu/IWO films on quartz substrates as a function of intermediate Cu thickness. It can be obtained that the resistivity of the bare IWO film is 9.59 × 10^−3^ Ω·cm, and the square resistance has reached 1.206 × 10^3^ Ω/□. However, the resistivity and square resistance of the multilayer thin film with a copper layer were greatly improved. The resistivity from 9.8 × 10^−4^ to 4.5 × 10^−4^ Ω·cm firstly decreases lightly then rapidly declines as the thickness grows from 4 to 10 nm. The changing trend is caused by changes in the state of the intermediate copper layer. According to the growth nucleation theory of metal thin film, in the initial growth stage the film is mainly composed of island-like particles, as shown in Figure 5a [42,43]. As the growth continues, the island-like clusters will grow and combine to form a semi-continuous porous network film as depicted in Figure 5b. Various microstructures of the metal film growth process correspond to different electrical conductivities. The conductivity at the island film stage is limited since the electron transport is generally through thermal and tunnel emissions [44]. The conductivity of the reticular film phase increases rapidly with the increase of the film thickness via the electrons transport between small metal islands, contact points or filaments, and inter-island gaps. The films are transitioning from non-metal to metal. When the thickness of the metal layer is <8 nm, the metal island boundary causes more scattering. The inherent disorder between these boundaries disperses the free carriers, so the space charge transport barrier is formed between the gaps, which requires a huge amount of external energy for the electrons to pass through. The conductivity model at this stage is given by the following model as in [40,44]:(5)$$\sigma \propto exp(-2\alpha L-W/k_{B}T)$$
where α represents the tunneling exponent of electron wave functions in the insulator, *W* denotes the island charging energy, *L* stands for the island separation, *k*_B_ is the Boltzmann constant, and *T* depicts the temperature.

When the Cu layer thickness is >8 nm, Cu layers evolve from an island shape to a porous semi-continuous morphology, which plays a fantastic role in the conduction mechanism of the IWO/Cu/IWO multilayer film. As the thickness increases, Cu islands are aggregated and merged, so the conductivity gradually approaches the intrinsic value, which is given by [45]:(6)$$\frac{\sigma }{\sigma_{0}}\propto \frac{3}{4}(1-p)\kappa log\frac{1}{\kappa }$$
(7)$$\kappa =d/\lambda_{0}$$
where *σ* denotes the conductivity of the metal film, σ_0_ is the bulk metal conductivity, λ_0_ stands for the mean free path of the electron, *d* represents the film thickness, and *p* depicts the surface reflection coefficient of the metal film.

The total resistance of the film is given by
(8)$$\frac{1}{R}=\frac{1}{R_{Cu}}+\frac{2}{R_{IWO}}$$
where *R_IWO_* denotes the resistance of the IWO layer and *R_Cu_* is the resistance of the intermediate Cu layer. It is obvious that the resistance value of the multilayer film can be reduced.

## 4. Conclusions

IWO/Cu/IWO multilayer structures were deposited on quartz substrates and characterized as low resistance transparent conductive films. Hall measurement data shows that the conductivity of the multilayer film is completely dependent on the thickness of the intermediate Cu layer, and the optimized film resistance is as low as 4.5 × 10^−4^ Ω·cm. The maximum optical transmittance in the mid-infrared region is up to 72%. Therefore, low-resistivity, mid-infrared transparent conductive films can be synthesized at room temperature without requiring a high substrate temperature, which serves as an excellent candidate for the mid-infrared transparent electrode.

## Figures and Tables

**Figure 1 materials-13-00113-f001:**
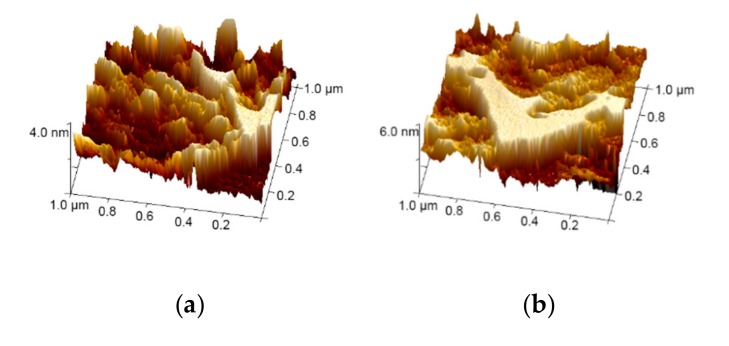
AFM images (1 μm × 1 μm) of the Cu layer with different thicknesses of (**a**) 4 nm, (**b**) 6 nm, and (**c**) 8 nm, deposited on the bottom IWO.

**Figure 2 materials-13-00113-f002:**
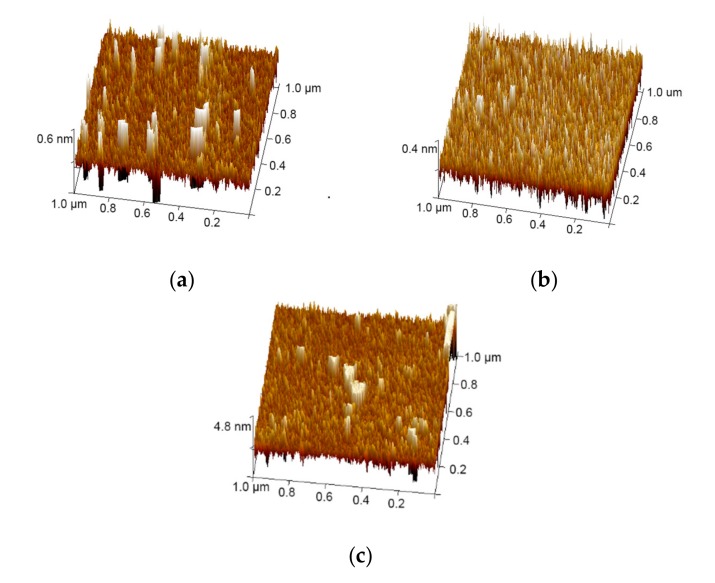
(**a**) AFM images (1 μm × 1 μm) of the bare IWO thin film. (**b**) AFM images (1 μm × 1 μm) of the bottom IWO on sapphire substrate. (**c**) AFM image (1 μm × 1 μm) of the multilayer film with a 6 nm Cu layer.

**Figure 3 materials-13-00113-f003:**
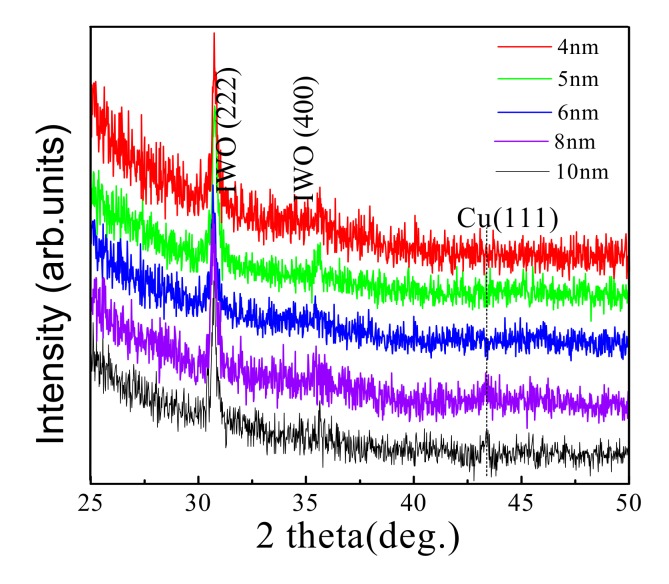
XRD patterns of five IWO/Cu/IWO films on quartz substrates. The thicknesses of the intermediate Cu are 4, 5, 6, 8, and 10 nm, respectively. The IWO film has an unchanged thickness of 40 nm.

**Figure 4 materials-13-00113-f004:**
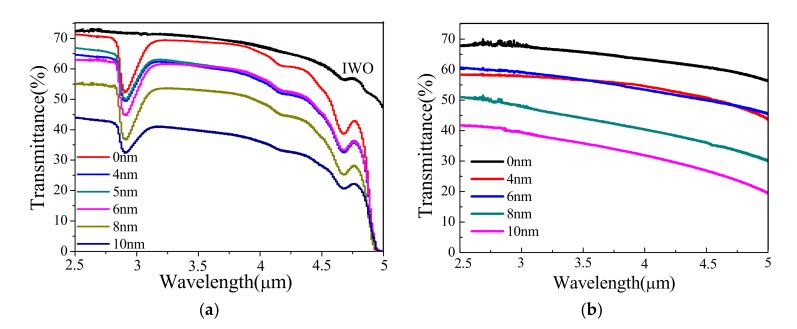
(**a**) Mid-infrared transmittance spectra of bare IWO thin film and IWO/Cu/IWO films on quartz substrates with Cu-layers of different thicknesses. (**b**) Mid-infrared transmittance spectra of IWO/Cu/IWO films on sapphire substrates as a function of Cu-layer thickness.

**Figure 5 materials-13-00113-f005:**
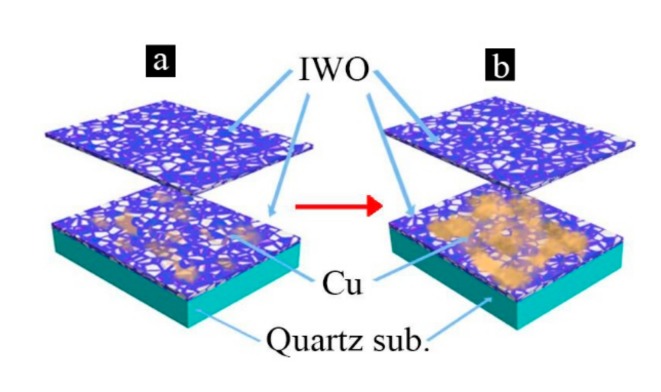
Schematic diagram of the evolution of the Cu layer in the IWO/Cu/IWO tri-layer films on quartz substrates. (**a**) Discontinuous Cu islands scatter on the IWO film. (**b**) Cu islands merge and form the semi-continuous porous layer.

**Figure 6 materials-13-00113-f006:**
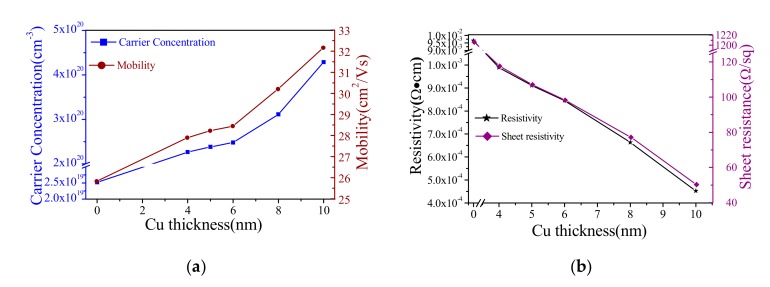
(**a**) Carrier mobility and concentration of IWO/Cu/IWO films with different Cu layer thicknesses on quartz substrates. (**b**) Resistivity and sheet resistance of bare IWO thin film and IWO/Cu/IWO films with various Cu layer thicknesses on quartz substrates.

**Table 1 materials-13-00113-t001:** Sputtering parameters for IWO and Cu layers.

Layer Type	Ar Flow (sccm)	Power	Growth Time	Growth Rate
IWO (upper)	25	35 W	50 min	0.8 nm/min
Cu	25	50 W	0, 1, 1.25, 1.5, 2, and 2.5 min	4 nm/min
IWO (bottom)	25	35 W	50 min	0.8 nm/min

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
