# Peer review of "Influence Mechanism of Cu Layer Thickness on Photoelectric Properties of IWO/Cu/IWO Films"

_materials, 2019, doi:10.3390/ma13010113_

Round 1

Reviewer 1 Report

In this paper, the Authors present the photoelectric properties of the IWO/Cu/IWO multilayer films as a function of Cu layer thickness. Below are some major issues that I recommend you to consider in further improving this manuscript.

* Authors stated that in the experimental section in line 79 “The electrical properties of the films were characterized by a SKP5050 Kelvin Probe system manufactured by KP Technology Ltd. in United 80 Kindom.”.

The Kelvin Probe (SKP5050 Kelvin Probe system) is a non-contact, non-destructive vibrating capacitor device used to measure the work function (wf) of conducting materials or surface potential of semiconductor or insulating surfaces. I was wondering how you have characterized the electrical properties of IWO film.

Need explanation?

* Surface morphology and surface roughness of IWO before and after Cu deposition should be added as the Cu film thickness is very low. The surface roughness of the IWO film might influence the morphology of Cu film. Also, Cu film morphology may have an impact of the final IWO film morphology. Details analysis should be added to the manuscript.

*Figure 1 scale bar is difficult to see. Remove the black background and provide better Figure 1.

*   The entire text should be carefully checked for language errors.

For instance, in Page 5, line 157, “Electrical properties” is misspelled.

*It is completely unclear how many samples were prepared for each Cu thickness, and no error analysis is reported for the electrical properties (carrier concentration, mobility, and resistivity). Error analysis and numbers of runs need to be provided throughout the paper.

*     I recommend that the comparison of values of resulting film properties with previously reported works prepared by different techniques should be added so the reader gets an idea on how much this work advances,

Author Response

Responses to Reviewer #1

We would like first to thank the reviewers for their comments, which was useful and helpful for us to improve both the quality and the clarity of our manuscript.

1. the Kelvin Probe (SKP5050 Kelvin Probe system) is a non-contact, non-destructive vibrating capacitor device used to measure the work function (wf) of conducting materials or surface potential of semiconductor or insulating surfaces. I was wondering how you have characterized the electrical properties of IWO film. Need explanation?

Reply: Thanks for your suggestions. The SKP5050 Kelvin Probe system we used is the SKP5050 Kelvin Probe and Hall Effect Tester. This instrument can measure the work function or surface potential of materials, and can also measure the Hall effect, magnetic resistance and I-V characteristics. The revised sentences for explanation are: “The thickness of the films was measured by a XP-2 step profiler (AMBIOS Trchnology Corporation, USA). The electrical properties of the films were characterized by a SKP5050 Kelvin Probe and Hall Effect Tester (KP Technology Ltd., United Kindom), which were used to measure the work function or surface potential of materials and to test the Hall effect and magnetoresistance characteristics.” (lines 87-90).

2. Surface morphology and surface roughness of IWO before and after Cu deposition should be added as the Cu film thickness is very low. The surface roughness of the IWO film might influence the morphology of Cu film. Also, Cu film morphology may have an impact of the final IWO film morphology. Details analysis should be added to the manuscript.

Reply: Thank you for your kind suggestion. We added the AFM image of a bare IWO film without Cu layer (Figure 2) and the AFM image of a multilayer film with a 6 nm Cu layer. Analysis of the roughness was also added.

“Surface morphology of the evolution progress obtained by atomic force micrographs (AFM) of Cu layers prepared on IWO films by RF magnetron sputtering are shown in Figure 1. The root-mean-square surface roughness of Cu layers was found to be (a) 1.36 nm, (b) 1.28 nm and (c) 2.12 nm, respectively. The growth morphology of Cu layer was systematically studied in order to evaluate the influence of the intermediate Cu layer on the photoelectric properties of the multilayer films. The evolution of the Cu layer is concluded as the following processes: separated Cu islands, mixed Cu islands and semi-continuous Cu layers. Figures 1(a) and 1(b) show the structure of the Cu island which can form semi-continuous layer as it reaches the critical thickness, as shown in Fig. 1(c). There is evident variation in root-mean-square surface roughness, which declines from 1.36 nm to 1.28 nm for the Cu layers with 4 nm and 6 nm thicknesses. This can be a result of the merger of smaller islands. As the Cu layer thickness is increased to 8 nm, the film shows great undulations and the roughness rises to 2.12 nm.

Figure 2 shows the surface morphology images of the bare IWO (80 nm thickness) thin film with a roughness of 0.27 nm, the IWO film with a roughness of 0.23 nm on a sapphire, and a multilayer film with a 6 nm Cu layer with a surface roughness of 1.29 nm. This shows that the substrate does not greatly affect the surface morphology of the IWO film. The surface topography in Figure 2 (b) indicates that, the bottom IWO film reveals uniform and smooth surface, which is critical for growing the Cu layer with a good surface topography. However, the surface roughness of the IWO/Cu/IWO film in Figure 2(c) is large because of the large roughness of the intermediate Cu layer, which affects the growth of the upper IWO film.”

Please refer to lines 102-121.

3. Figure 1 scale bar is difficult to see. Remove the black background and provide better Figure 1

Reply: Thanks for your suggestions. We removed the black background and changed it to a white background.

   Please refer to Figure 1 on Page 4.

4. The entire text should be carefully checked for language errors. For instance, in Page 5, line 157, “Electrical properties” is misspelled 

Reply: Thank for your suggestions. We corrected some errors and double-checked the language of our manuscript carefully.

5. It is completely unclear how many samples were prepared for each Cu thickness, and no error analysis is reported for the electrical properties (carrier concentration, mobility, and resistivity). Error analysis and numbers of runs need to be provided throughout the paper.

Reply: Thanks for your suggestions. We added the number of runs of the experiment and explained the error of the experimental sample thickness and electrical properties (carrier concentration, mobility, and resistivity).

“different IWO films were grown on the substrate, whose thicknesses were determined by sputtering time. With a step profiler, the measurement results of more than 10 groups of samples prove that the deposition rate can be accurately controlled with an error of < 5%. The growth rate of the Cu layer on the IWO film is also stable. Finally, the obtained samples were annealed at 350℃ for good electrical performances.” (lines 76-81)

“Figure 6 reveals the variation in the carrier concentration, mobility, resistivity and sheet resistance of the IWO/Cu/IWO thin films as a function of Cu layer thickness. The morphology of the copper layer is related to the surface morphology and roughness of the underlying IWO film, which directly affects the electrical properties of the multilayer film. Here, we performed each group of experiment more than twenty times to determine the error of the electrical properties of the thin film by analyzing the obtained carrier concentration, mobility and resistivity. Although the results obtained are slightly different due to the deviation of Cu, the error is < 6%. The resistivity of the bare IWO thin film is 9.86×10-3 Ω·cm, which is basically the same as that of the multilayer film without a Cu layer. Figure 6(a) shows the relationship of carrier concentration and Hall mobility of IWO/Cu/IWO films. With increasing Cu layer thickness, an obvious improvement in Hall mobility is observed from 25.8 cm2/Vs at 0 nm to 32.1 cm2/Vs at 10 nm. The carrier concentration is increased significantly from 2.52×1019 to 4.28×1020 cm-3, which depends on the increase of the thickness of the Cu layer. Hence, the conduction of Cu is dominant in this multilayer system. As shown in Figure 6(a), for the IWO film without a Cu layer, the carrier concentration is 2.52×1019 cm-3 only; furthermore, with a Cu layer, the carrier concentration rises sharply.” (lines 185-191)

6. I recommend that the comparison of values of resulting film properties with previously reported works prepared by different techniques should be added so the reader gets an idea on how much this work advances. 

Reply: Thank for your suggestions. We added a discussion of some related works to the revised manuscript: “Some multilayer films composed of a variety of TCO and metals have been reported using different high-temperature deposition methods, such as electron beam evaporation [27,28], sputtering [29], chemical vapor deposition [30], and pulsed laser deposition [31]. Magnetron sputtering method is a valid technique that can produce films with reasonable quality at a stable deposition rate. However, mid-infrared photoelectric properties of multilayer films deposited by magnetron sputtering at room temperature have rarely been reported.” (lines 43-49)

Reviewer 2 Report

The paper is very interesting, but I have one minor style remark. The second sentence in Introduction (lines 20-23) is rather hard for understand. It is desirable to divide it.

Transparent conducting oxides with very thin metal film within TCO layer have recently received a renewed interest as a promising route in the framework of very thin devices. WIO layers, used in the paper, are transparent in the mid-IR range and less investigated than typical TCOs (doped zink and tin oxides). Low resistivity and high transmittance in the WIO/Cu/WIO films have been obtained. These values are the same as published data for other visibly transparent films. The multilayer films were fabricated at room temperature. Such technology is rarely reported. WIO/Cu/WIO films might be rather advantageous systems for the mid-IR range application.

Author Response

The second sentence in Introduction is rather hard for understand. It is desirable to divide it.

Reply: The sentence was revised: “TCOs usually consist of transition metal cations (called TCO cations) because the powerful delocalized s orbitals of these cations form a dispersed conduction band. Their effective electron mass is very small which assures high mobility and wide band gap” (lines 20-23)

Reviewer 3 Report

In the manuscript authors prepared IWO ( W-doped In2O3 )/Cu/IWO sandwich thin film structure by magnetron sputtering at room temperature and optimized their electrical and optical properties in mid-infrared range by variation of Cu layer thickness for application as a superior mid-infrared transparent electrode. As the main novelty of the manuscript, they stated the optimization of mid-infrared optical properties.

The manuscript is well structured and the content is expressed clearly and in proper English except several minor typing errors. But as main weakness, I see the fact that authors missed to compare electrical and optical properties of prepared sandwich structures with bare IWO thin film. By adding results also for bare IWO layer they will support significantly stronger their conclusions about superior electrical and optical properties of the sandwich structure. Also, the optical measurement was done and presented for samples prepared at quartz and sapphire substrates but AFM and electrical characterization not. I’m also suggesting to add also results for AFM and Hall, in the same way, to support the conclusion that there is no substrate influence on surface morphology and electrical properties.

There are also other parts that should be clarified in more details:

Sample preparation is Deposition rates for IWO and Cu were stated in the experimental part but it’s not clear how they are determined? Monitored by quartz crystal microbalance?

In Figure 1 it’s not clear how are distinguished Cu islands from IWO surface? For clarity, they should be labelled. Also, additional AFM image of bare IWO would help in that sense.

In Results are mentioned postdeposition annealing at 350 C deg. but not described in the Experimental part. What is the purpose and results of this step?

As mentioned before, in Fig. 3 and Fig. 4 it would be interesting to compare results with bare IWO layer.

Eq 2. It’s not clear what is epsilon_m

Fig. 6: as for Fig 3 would be interesting to discuss difference to bare IWO layer

Author Response

Responses to Reviewer #3

We would like first to thank the reviewers for their comments, which was useful and helpful for us to improve both the quality and the clarity of our manuscript.

 1. I see the fact that authors missed to compare electrical and optical properties of prepared sandwich structures with bare IWO thin film. By adding results also for bare IWO layer they will support significantly stronger their conclusions about superior electrical and optical properties of the sandwich structure. Also, the optical measurement was done and presented for samples prepared at quartz and sapphire substrates but AFM and electrical characterization not. I’m also suggesting to add also results for AFM and Hall, in the same way, to support the conclusion that there is no substrate influence on surface morphology and electrical properties.

Reply: We added the AFM image of the bare IWO film and the IWO film on the substrate, as shown in Figure 2. The roughness analysis shows that the substrate has no effect on the surface morphology of the IWO film. A comparison of the optical characteristics of the bare IWO thin film was added in Figure 4 to illustrate the optical characteristics of the multilayer structure film. In addition, we added the electrical properties of the bare IWO film in Figure 6. The electrical properties of the IWO film and the multilayer film were discussed as a comparison, and the substrate had no effect on the electrical properties. We discussed the comparability of the bare IWO film and the multilayer in detail and added them to corresponding chapter in the revised manuscript:

“Figure 2 shows the surface morphology images of the bare IWO (80 nm thickness) thin film with a roughness of 0.27 nm, the IWO film with a roughness of 0.23 nm on a sapphire, and a multilayer film with a 6 nm Cu layer with a surface roughness of 1.29 nm. This shows that the substrate does not greatly affect the surface morphology of the IWO film. The surface topography in Figure 2 (b) indicates that, the bottom IWO film reveals uniform and smooth surface, which is critical for growing the Cu layer with a good surface topography. However, the surface roughness of the IWO/Cu/IWO film in Figure 2(c) is large because of the large roughness of the intermediate Cu layer, which affects the growth of the upper IWO film.” (lines 114-121)

“The transmission spectra in the mid-infrared region (2.5-5 μm) were obtained from the deposited bare IWO thin film and multilayer films. It is evident that the transmittance of the multilayer structure is lower than that of the bare IWO film due to the embedded Cu layer, reaching a maximum of 72%. As shown in Figure 4(a), the absorption peak at 2.73 μm is due to the hydroxyl impurity inside the quartz substrate. With the increase of the Cu layer thickness from 4 nm to 6 nm, the transmittance increases first and then decreases in the region of 2.5-3.75 μm. The maximum transmittance of 67% in 2.5-5 μm is attained on IWO/Cu/IWO multilayer films with a 5-nm thick Cu layer. Figure 4(b) shows the transmission spectra of IWO/Cu/IWO films on sapphire substrates in the mid-infrared region (2.5-5 μm). It is clearly that the transmission also increases first and then decreases.” (lines 140-149) 

“We performed each group of experiment more than twenty times to determine the error of the electrical properties of the thin film by analyzing the obtained carrier concentration, mobility and resistivity. Although the results obtained are slightly different due to the deviation of Cu, the error is < 6%.” (lines 186-189)

2. Sample preparation is Deposition rates for IWO and Cu were stated in the experimental part but it’s not clear how they are determined? Monitored by quartz crystal microbalance?

Reply: We deposited films through different time gradients, and then used a step meter to measure the film thickness and determined the film deposition rate by calculation. The deposition rate is very stable when the experimental conditions are determined, which is consistent with the data measured by the step meter for the films prepared.

“Here, different IWO films were grown on the substrate, whose thicknesses were determined by sputtering time. With a step profiler, the measurement results of more than 10 groups of samples prove that the deposition rate can be accurately controlled with an error of < 5%. The growth rate of the Cu layer on the IWO film is also stable. Finally, the obtained samples were annealed at 350℃ for good electrical performances.” (lines 77-81)

3. In Figure 1 it’s not clear how are distinguished Cu islands from IWO surface? For clarity, they should be labelled. Also, additional AFM image of bare IWO would help in that sense. 

Reply: We labeled the Cu layer and IWO layer in Figure 1.

4. In Results are mentioned postdeposition annealing at 350 C deg. but not described in the Experimental part. What is the purpose and results of this step? 

Reply: The revised sentence is: “Finally, the obtained samples were annealed at 350℃ for improving the electrical performances.” (lines 80-81)

5. As mentioned before, in Fig. 3 and Fig. 4 it would be interesting to compare results with bare IWO layer.

Reply: We added the transmission spectra of the bare IWO film.

“The transmission spectra in the mid-infrared region (2.5-5 μm) were obtained from the deposited bare IWO thin film and multilayer films. It is evident that the transmittance of the multilayer structure is lower than that of the bare IWO film due to the embedded Cu layer, reaching a maximum of 72%. As shown in Figure 4(a), the absorption peak at 2.73 μm is due to the hydroxyl impurity inside the quartz substrate. With the increase of the Cu layer thickness from 4 nm to 6 nm, the transmittance increases first and then decreases in the region of 2.5-3.75 μm. The maximum transmittance of 67% in 2.5-5 μm is attained on IWO/Cu/IWO multilayer films with a 5-nm thick Cu layer. Figure 4(b) shows the transmission spectra of IWO/Cu/IWO films on sapphire substrates in the mid-infrared region (2.5-5 μm). It is clearly that the transmission also increases first and then decreases.” (lines 140-149) 

6. Eq 2. It’s not clear what is epsilon_m 

Reply: We explained it in the text: “where εm denotes the dielectric constant of the bulk metal.” (line 161)

7. Fig. 6: as for Fig 3 would be interesting to discuss difference to bare IWO layer

Reply: We added the electrical properties (carrier concentration, mobility, and resistivity) of the bare IWO film in Figure 6. The electrical characteristics of the bare IWO film and the multilayer structure film were compared and discussed in detail. The relative contents are presented as follows:

“Figure 6 reveals the variation in the carrier concentration, mobility, resistivity and sheet resistance of the IWO/Cu/IWO thin films as a function of Cu layer thickness. The morphology of the copper layer is related to the surface morphology and roughness of the underlying IWO film, which directly affects the electrical properties of the multilayer film. We performed each group of experiment more than twenty times to determine the error of the electrical properties of the thin film by analyzing the obtained carrier concentration, mobility and resistivity. Although the results obtained are slightly different due to the deviation of Cu, the error is < 6%. The resistivity of the bare IWO thin film is 9.86×10-3 Ω·cm, which is basically the same as that of the multilayer film without a Cu layer. Figure 6(a) shows the relationship of carrier concentration and Hall mobility of IWO/Cu/IWO films. With increasing Cu layer thickness, an obvious improvement in Hall mobility is observed from 25.8 cm2/Vs at 0 nm to 32.1 cm2/Vs at 10 nm. The carrier concentration is increased significantly from 2.52×1019 to 4.28×1020 cm-3, which depends on the increase of the thickness of the Cu layer. Hence, the conduction of Cu is dominant in this multilayer system. As shown in Figure 6(a), for the IWO film without a Cu layer, the carrier concentration is 2.52×1019 cm-3 only; furthermore, with a Cu layer, the carrier concentration rises sharply.” (line 183-197)

“Figure 6 (b) shows the volume resistivity and sheet resistance of IWO/Cu/IWO films on quartz substrates as a function of intermediate Cu thickness. It can be obtained that the resistivity of the bare IWO film is 9.59×10-3 Ω·cm, and the square resistance has reached 1.206×103 Ω/□. However, the resistivity and square resistance of the multilayer thin film with copper layer are greatly improved.” (line 215-218)

Round 2

Reviewer 1 Report

The authors have properly addressed the majority of the comments from the reviewers. I think the revised manuscript can be accepted for publication.

Author Response

No revision needed in response to the comment of this reviewer.

Reviewer 3 Report

The authors have taken the reviewers' comments seriously and revised the manuscript accordingly. In my opinion, the revised manuscript could be considered for publication in the Coatings journal.
However, the transmission data for bare IWO film added in Figure 4a could be possibly explained in more details. Compared to IWO/Cu/IWO sandwich films does not have an absorption peak at 2.73 μm which is explained as a contribution from quartz substrate. Does it mean that bare IWO is not deposited on quartz substrate? Or there is another explanation?

Author Response

Comment: I see the fact The transmission data for bare IWO film added in figure 4a could be possibly explained in more details. Compared to IWO/Cu/IWO sandwich films does not have an absorption peak at 2.73 um which is explained as a contribution from quartz substrate. Does it mean that bare IWO is not deposited on quartz substrate? Or there is another explaination?

Reply: The transmission spectrum of the IWO thin film on quartz substrate was corrected by that of the uncovered quartz substrate, and then we can obtain the transmission spectrum of the bare IWO thin film without the absorption peak at 2.73 μm. (Line 146-148)